# Low-Pass Filtering Improves Behavioral Alignment of Vision Models

**Max Wolff**[*1,2]  **Thomas Klein**[*1,2,3,4]  **Evgenia Rusak**[1,2,3,4,5]

**Felix A. Wichmann**[‡2]  **Wieland Brendel**[‡1,2,3,4]

## Abstract

Despite their impressive performance on computer vision benchmarks, Deep Neural Networks (DNNs) still fall short of adequately modeling human visual behavior, as measured by error consistency and shape bias. Recent work hypothesized that behavioral alignment can be drastically improved through *generative*—rather than *discriminative*—classifiers, with far-reaching implications for models of human vision.

Here, we instead show that the increased alignment of generative models can be largely explained by a seemingly innocuous resizing operation in the generative model which effectively acts as a low-pass filter. In a series of controlled experiments, we show that removing high-frequency spatial information from discriminative models like CLIP drastically increases their behavioral alignment. Simply blurring images at test-time—rather than training on blurred images—achieves a new state-of-the-art score on the model-vs-human benchmark, halving the current alignment gap between DNNs and human observers. Furthermore, low-pass filters are likely optimal, which we demonstrate by directly optimizing filters for alignment. To contextualize the performance of optimal filters, we compute the frontier of all possible pareto-optimal solutions to the benchmark, which was formerly unknown.

We explain our findings by observing that the frequency spectrum of optimal Gaussian filters roughly matches the spectrum of band-pass filters implemented by the human visual system. We show that the contrast sensitivity function, describing the inverse of the contrast threshold required for humans to detect a sinusoidal grating as a function of spatiotemporal frequency, is approximated well by Gaussian filters of the specific width that also maximizes error consistency.

## 1 Introduction

While Deep Neural Networks (DNNs) are widely considered the best models of the human visual system (Kriegeskorte, 2015; Kietzmann et al., 2017; Cichy & Kaiser, 2019; Doerig et al., 2023), there is a large body of research exposing the many ways in which DNNs exhibit non-human-like behavior—see (Wichmann & Geirhos, 2023) for an overview.

One behavioral dimension on which DNNs differ from humans is their lack of shape bias: Geirhos et al. (2019); Baker et al. (2018) show that when presented with stimuli characterized by conflicting shape- and texture cues, humans will classify images according to the shape cue, while DNNs will classify according to the texture cue. Furthermore, DNNs systematically disagree with human observers about which images they find difficult, as Geirhos et al. (2021) show using the *error consistency* metric (Geirhos et al., 2020; Klein et al., 2025). Both of these findings indicate that humans and DNNs likely implement different strategies for solving the task of core object recognition (DiCarlo et al., 2012), raising concerns about their suitability as computational models of the human

---

[*]Equal contribution. [‡] Joint supervision.
[1] Max Planck Institute for Intelligent Systems, Tübingen, Germany. [2] University of Tübingen, Germany. [3] Tübingen AI Center. [4] ELLIS Institute Tübingen. [5] Cohere.
Correspondence to: m.wolff1621@gmail.com, t.klein@uni-tuebingen.de.

visual system. Progress towards behavioral alignment of DNNs as measured by shape bias, error consistency and out-of-distribution (OOD) robustness is monitored by the `model-vs-human` (MvH) benchmark (Geirhos et al., 2021).

Previous state-of-the-art results had been achieved by very large models trained on gigantic, diverse datasets (CLIP (Radford et al., 2021) and ViT-22B (Dehghani et al., 2023)). However, Jaini et al. (2023) have recently shown that generative models such as Imagen (Saharia et al., 2022) achieve the most human-like behavior to date, with an error consistency of $0.31$ and a shape bias of $0.99$. Interestingly, Imagen also exhibits a bias towards low-frequency features compared to other DNNs. Jaini et al. (2023) speculate that this tendency could be caused by the diffusion noise encountered during training, or the generative objective itself.

The hypothesis that the generative objective is the driver of improved shape bias and error consistency ties into a much larger debate about the necessity of generative models for robust vision: It is an open question whether the human visual system operates in a discriminative, bottom-up fashion (vision as inverse inference, (Von Helmholtz, 1867)), or whether it is a model-based system with top-down priors, e.g. for predictive coding (Yuille & Kersten, 2006). If Imagen were so human-like because of its generative objective, it could be a hint that the human visual cortex also relies on such principles.

However, a thus far neglected processing step in Imagen is its downscaling of inputs to a resolution of $64 \times 64$ pixels, rather than processing the full $224 \times 224$ pixel resolution. This downscaling operation is effectively a low-pass filter. We thus consider a different hypothesis, derived from our knowledge of the human visual system: Imagen's low-pass filtering might be the true reason for its increased human-like behavior, since both the optics of the human eye and early neural processing stages act as low-pass filters, at least at the short presentation times used in the MvH benchmark. To investigate this hypothesis, we test how low-pass filtering of input images *at test time* affects behavioral alignment, which is in contrast with earlier work that investigated the effects of training on low-pass filtered images.

We find that low-pass filtering images at test time can drastically improve the behavioral alignment between almost all models and human observers, as demonstrated in Figure 1. A ViT-H-14 OpenCLIP model tested on blurred images ($\sigma = 2.5$ px) achieves an error consistency of $0.37$, substantially surpassing that of Imagen ($0.31$) (Jaini et al., 2023), halving the remaining gap between human-DNN and human-human alignment (see Table 1).

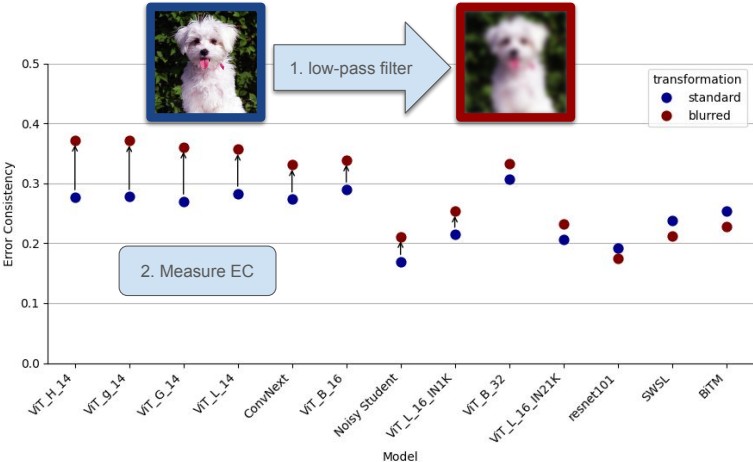

Figure 1: **Low-pass filtering images increases human-machine Error Consistency.** For the majority of investigated models, we find that low-pass filtering images prior to model evaluation can substantially increase their error consistency with human observers.

Prepending vision models with low-pass filters has a clear physiological motivation: Like any optical system, the human eye acts as a low-pass filter for the light hitting the retina (Campbell & Green, 1965; Williams et al., 1994), and the processing by the visual cortex acts as an additional filter over spatial frequencies (Campbell & Robson, 1968; Kelly, 1979). This filter is described by the contrast sensitivity function (CSF). The tuning curve of the CSF depends on the temporal frequency of the

signal, with higher temporal frequencies (i.e. shorter presentation times) moving the peak of the spectrum towards lower spatial frequencies (Kelly, 1979), turning the band-pass filter into a low-pass filter at very high temporal frequencies. We show that at the fairly short presentation times employed by MvH (200 ms), the CSF is approximated relatively well by low-pass filters, and the Gaussian that produces the highest error consistency to humans is close to the optimum approximation.

Furthermore, we observe that there is an inherent trade-off between accuracy and error consistency in MvH: To achieve maximal error consistency with human observers, a classifier has to roughly match human accuracy on the benchmark (see Klein et al. (2025)). This means that while the three metrics that make up model-vs-human—error consistency, shape bias and OOD-accuracy—have clear ceilings individually, the mathematical relationship between the metrics renders the overall ceiling score obtuse: It is not obvious what the best possible score on the benchmark is. We solve this problem by computing the frontier of all pareto-optimal solutions to the benchmark, thereby establishing the benchmark ceiling and revealing that MvH is not yet saturated, and much more progress can theoretically be made on achieving behavioral alignment between humans and machines.

Our contributions are thus the following:

1. We show that low-pass filtering images at test time (equivalent to blurring or resizing images) drastically increases a models' behavioral alignment as measured by error consistency and shape-bias, more so than training on low-pass filtered images.

2. We thus provide an alternative explanation for the increased behavioral alignment of Imagen, which was previously attributed to its generative objective.

3. We explain these findings by showing that the best low-pass filters approximate the CSF at short presentation times reasonably well, suggesting that low-pass filtering works because it matches the filtering implemented by the human visual system.

4. We analyze the model-vs-human benchmark, revealing an inherent trade-off between the two objectives of OOD-robustness and error consistency. We quantify this relationship by computing the frontier of all pareto-optimal solutions to the benchmark.

## 2 RELATED WORK

**DNNs as models of the human visual system.** The question of whether DNNs are suitable models of the human visual system has taken center stage of vision science research (Wichmann & Geirhos, 2023; Doerig et al., 2023; Bowers et al., 2022; Cichy & Kaiser, 2019; Schrimpf et al., 2018). Since a core property of any good model is its ability to reproduce the behavior of the target system, methods for measuring (and potentially improving) the behavioral alignment to humans are needed (Sucholutsky et al., 2023). Different approaches to this problem have been explored, e.g., Linsley et al. (2018); Muttenthaler et al. (2023), but at the center of our work is the model-vs-human benchmark by Geirhos et al. (2021). While various attempts have been made to increase shape bias (Geirhos et al., 2019; Li et al., 2020; Nuriel et al., 2021; Brochu, 2019), direct optimization towards increased error consistency is underexplored.

**Frequency tuning in the visual system.** The seminal work by Campbell & Robson (1968) gave rise to the idea that the early visual cortex is composed of *channels* sensitive to specific bandwidths, by noting that human contrast sensitivity is a function of spatial frequency, rather than global contrast alone. There is ample physiological evidence supporting this theory, e.g. De Valois et al. (1982). The emergence of human-like contrast sensitivity functions in DNNs has been explored as well (Li et al., 2022; Akbarinia et al., 2023), providing empirical evidence that DNNs exhibit human-like CSFs. Notably, these works find that DNN-CSFs are similar to the standard human CSF at unlimited viewing time. Schyns & Oliva (1994) showed that the channels of the human visual system do not operate at the same timescale: Humans process the low spatial frequencies first. Interestingly, this ordering (low to high) is also the order in which DNNs learn to make use of frequencies as features (Rahaman et al., 2019) (with more robust models relying on lower frequencies more (Li et al., 2023; Yin et al., 2019)), and there seem to be benefits of a visual diet that gradually introduces higher spatial frequencies (Vogelsang et al., 2018; 2024). Consequently, various attempts have been made to systematically train DNNs on low-pass filtered images (Jinsi et al., 2023; Jang & Tong, 2024), with one such training approach even achieving competitive shape bias results on MvH Lu et al.

(2025). Jang & Tong (2024) also achieve slightly higher error consistency with blur-trained models than standard ones, but their EC values stay well below $0.2$. In general, how DNNs utilize spatial frequencies is an active area of research: Subramanian et al. (2023) found that the bandwidth of frequencies to which DNNs are sensitive is much wider than that of humans, which is a potential source of behavioral differences between the two kinds of systems. They propose that narrowing the critical band of networks should make them more robust—in line with our finding that band-pass filtering to a "human-matched" frequency range increases alignment to humans.

## 3  METHODS

### 3.1  ALIGNMENT METRICS

To measure a model's behavioral consistency with humans, we use the `model-vs-human`[1] package (Geirhos et al., 2021). This benchmark consists of images from the training set of ImageNet-1k (Russakovsky et al., 2015) grouped into 16 coarse "super-classes" (airplane, bear, bicycle, bird, boat, bottle, car, cat, char, clock, dog, keyboard, knife, oven, truck) which subsume multiple of the more fine-grained ImageNet-1k classes at the basic level of classification (Rosch et al., 1976). The images are corrupted by 12 different parametric image distortions, such as additive noise or color inversion. In addition, the benchmark contains stylized, edge-filtered, silhouette, cue-conflict (see **Shape Bias** below) images and sketches. All of these images were shown to multiple human subjects, who had to solve a speeded classification task: Images were shown for 200ms, followed by a pink-noise backward mask, to suppress recurrent processing in the visual cortex as much as possible. Humans then had to classify images into one of the 16 super-classes. To evaluate a new model on the benchmark, the model first has to classify all images, so that three metrics measuring the similarity of the model to human judgments can be computed, which we explain next.

**Shape Bias.**   The shape bias metric is computed on the cue-conflict images. These images were generated using neural style transfer (Gatys et al., 2015) by combining the content (shape) of one image with the texture (style) of another. The dataset consists of 1200 images, sampled evenly from the 16 classes. To calculate shape bias, the model is first evaluated on all cue-conflict images. Then, all trials are discarded on which neither the class implied by the shape cue, nor the class implied by the texture cue was predicted. Shape bias is then defined as the proportion of the remaining trials for which the model decides according to the shape cue. Notably, the metric does not take a model's overall accuracy into account, meaning that one can achieve high shape bias by classifying a few images into the class implied by the shape cue and predicting nonsense on other images (Doshi et al., 2024). Shape bias takes on values between 0 and 1, with higher shape bias indicating higher behavioral alignment, because humans exhibit almost perfect shape bias (0.96).

**Error Consistency.**   This metric indicates the extent to which two decision makers (in this case, a model and a human observer) make errors on the same images (Geirhos et al., 2020; 2021). It is measured on all 17 experiments from the `model-vs-human` package, excluding a handful of conditions on which humans performed below a threshold. Error consistency is computed by calculating Cohen's $\kappa$ (Cohen, 1960) on two binary sequences indicating whether each classifier gave a correct response on a trial, see Geirhos et al. (2020); Klein et al. (2025) for details. $\kappa$ takes on values between -1 and 1, with 1 indicating maximum agreement, 0 meaning that they agree not more than expected by chance, and -1 indicating maximum disagreement. Within the model-vs-human benchmark, EC values are averaged in a hierarchical fashion, by first averaging across humans within each corruption strength, then across corruption strengths, and finally across corruptions. While a higher EC to humans is always indicative of more behavioral alignment, it is not necessarily possible for EC to take on all values in $[-1, 1]$, since $\kappa_{max}$ depends on the accuracy mismatch between the classifiers (see Klein et al. (2025) for an in-depth explanation).

**OOD Accuracy.**   This metric measures the aggregate accuracy of a classifier on all 17 datasets. Higher OOD-accuracy is considered better, because humans typically demonstrate higher OOD-accuracy on the model-vs-human corruptions. (Note that recently progress in model robustness has

---

[1] https://github.com/bethgelab/model-vs-human

increased to the point of a paradigm shift, where models now outperform humans on some corruptions (Li et al., 2025).)

## 3.2 MODELS AND TRANSFORMATIONS

We evaluate models trained on LAION-2B available through the open-source OpenCLIP package (Schuhmann et al., 2022; Radford et al., 2021; Cherti et al., 2023; Ilharco et al., 2021). In our experiments, we evaluate CLIP in the zero-shot setting using the 16 `model-vs-human` classes and the standard 80 prompt averaging scheme.

For blur and resize transformations, we use `torchvision` (Marcel & Rodriguez, 2010; Paszke et al., 2017) implementations of `GaussianBlur` and `Resize` with bi-cubic interpolation. We resize the image from its original size of $R_0 \times R_0$ to the target resolution $R_1 \times R_1$, and back up to $R_0 \times R_0$. This emulates the resizing that Imagen conducts. A higher "resize strength" means resizing to a lower resolution, i.e. a larger ratio $R_0/R_1$. We visualize the strengths of blur and resize transformations that we use in Figure 7.

## 3.3 LEARNING A FOURIER FILTER TO MAXIMIZE ERROR CONSISTENCY

We also experiment with learning a Fourier filter that maximizes error consistency with humans. To do so, we use the `model-vs-human` images and corresponding human predictions, and the frozen OpenCLIP ViT-H-14 model. We first compute an optimal binary vector of human responses, as outlined in Appendix A.3. Our loss then induces the model to give classification responses leading to the same correctness values, by essentially optimizing the cross-entropy between the optimal correctness values and the model correctness values.

If the ideal response is the correct label, we induce the model to predict the correct label as well. Otherwise, we instead induce the model to predict the incorrect class which currently seems most likely to the model, because this particular wrong response (out of the 15 possible wrong responses) should be easiest to learn.

Since our filter $G_\theta$ needs to respect Hermitian symmetry and we only want to modulate frequencies without conducting phase shifts, we parameterize the $224 \times 224$ DFT-matrix of the filter as a real $112 \times 112$ matrix. (Note that most of the MvH images are in grayscale; we otherwise apply the filter to each color channel independently.) These parameters form the top left quadrant of the filter, and we obtain the remaining quadrants by computing the complex conjugates, which of course still all have an imaginary component of zero. Since we are limited to the $< 12,000$ images in MvH to learn $112 \times 112 = 12,544$ parameters, optimizing this filter directly would cause overfitting and lead to a noisy "adversarial" pattern. To prevent overfitting, we regularize the filter to enforce sparsity and smoothness, and use all available images for training, without a train-test split. Sparsity is enforced via an $L^1$ regularization over filter parameters, while smoothness is achieved by blurring the filter itself. Let $F$ be the Fourier transform, $f_I$ be the image encoder, and $f_T$ the text encoder of the CLIP model. Let $x$ be an image, and $G_\theta$ the filter to be learned. Let $b_\gamma$ be a Gaussian blur function, parameterized by the blur strength $\gamma$. Then, each element $s_i$ of the vector $s$ of cosine similarities between the image $x$ and all "labels" $y_i$ is given by

$$s_i = sim\bigg(f_I\big(F^{-1}(F(x) * b_\gamma(G_\theta))\big), f_T(y_i)\bigg). \tag{1}$$

Let $\hat{y}$ denote the one-hot encoded ground-truth label of $x$, $H$ the cross-entropy, and $\sigma$ the softmax function. Then the loss for a single image is given by

$$\mathcal{L}(x) = H\bigg(\sigma(\frac{s}{\tau}), \hat{y}\bigg) + \lambda\|\theta\|_1. \tag{2}$$

Since CLIP similarities are too close in magnitude to provide enough signal to the cross-entropy loss (after Softmax), we introduce a learned temperature parameter $\tau$, similar to Radford et al. (2021). Using a grid search, we found that an $L^1$ weight of $5 \times 10^{-5}$ and a blurring strength of $\gamma = 6.0$ yielded the best error consistencies with humans. We use the Adam optimizer, and initialize the filter from ones with a small amount of Gaussian noise ($\sigma^2 = 10^{-5}$) added.

Once we have obtained the final filter, we apply it to the `model-vs-human` images, classify them using the frozen model, and evaluate the error consistency with humans (exposed to clean images).

### 3.4 Evaluating goodness of fit to the CSF

The contrast sensitivity function (CSF) characterizes the contrast sensitivity of human observers as a function of spatial and temporal frequency (Kelly, 1979). We use the empirical estimates of the CSF obtained by Kelly (1979), which we describe in more detail in Appendix A.2. To quantify the goodness of fit between the human CSF (normalized to the range $[0, 1]$) and a Gaussian of a specific $\sigma$, we compute $\mathcal{L}_{WRMSE}$ between the normalized CSF and the Gaussian, for the relevant frequency range. This range is limited by the display apparatus in model-vs-human, where images were presented at $3°$ of visual angle, so $f_{min} = \frac{1}{3}$ cycles per degree, and the Nyquist frequency of the monitor, which was $f_{max} = 42.58$ cpd. To take into account that the spectrum of natural images is roughly $f^{-1}$ and that the power of the spectrum is therefore $P(f) = f^{-\beta}$, with $\beta \approx 2$, we weight the errors accordingly:

$$\mathcal{L}_{WRMSE} = \sqrt{\frac{\int_{f_{min}}^{f_{max}} f^{-\beta}(CSF(f) - G(f))^2 \, df}{\int_{f_{min}}^{f_{max}} f^{-\beta} \, df}}. \tag{3}$$

The 1D spectrum depicted in Figure 4 can be thought of as the cross-section of a radially symmetric filter, thus giving rise to a 2D spectrum that can be scaled appropriately and applied to images.

## 4 Results

### 4.1 Low-pass filtering explains Imagen's alignment.

We hypothesize that Imagen outperforms our reference model on error consistency (EC, the degree to which a model makes the same mistakes as humans) and shape bias (SB, the tendency of a model to prefer shape cues over texture cues) simply because of its lower input resolution of $64 \times 64$ pixels, which amounts to low-pass filtering of the input. To test this hypothesis, we prepend the reference model with a low-pass filter, which we implement once as a resizing operation analogous to Imagen, and once as a Gaussian blur, which should have a very similar effect. We vary the blurring strength and observe the resulting changes in performance on the three alignment metrics. As a reference model, we choose OpenCLIP ViT-H-14, because it is the most well-aligned model we have access to, with a baseline EC of $0.28$, an SB of $0.60$ and an OOD-accuracy of $0.78$.

**Shape Bias.** Increasing either blur- or resize-strength monotonically increases shape bias (see Figure 2). Since blurring removes high-frequency texture information, the model is forced to rely on low-frequency shape cues. Independent of the method of implementation, low-pass filtering of input images alone can account for Imagen's increased shape bias, even in models without the generative component that was formerly hypothesized to explain Imagen's human-level shape bias.

**Error Consistency.** As we increase the blurring strength, the model's EC rises from $\kappa = 0.28$ to $\kappa = 0.37$ (after blurring with $\sigma = 2.5$) or $\kappa = 0.35$ (after resizing to $64 \times 64$), respectively (see Figure 2). For reference, the previous highest error consistency had been reported for Imagen (Saharia et al., 2022) at $\kappa = 0.31$. After a certain "critical point," however, error consistency begins to decrease for higher blurring/resizing strengths. We break down the error consistency gains into the different MvH conditions in Appendix A.5, but observe gains across conditions.

**OOD Accuracy.** Notably, these transformations do have a significant effect on the model's OOD accuracy: OOD accuracy dips by 6 percentage points after blurring at $\sigma = 2.5$ and by 3 percentage points when resizing the image to $64 \times 64$ (see Table 1). This is to be expected, since we are effectively removing features that the model can use. As we argue in Section 4.5, this drop in accuracy may actually be necessary to improve behavioral alignment. However, the drop in accuracy is by no means catastrophic. This is important, because shape bias does not account for accuracy (Doshi et al., 2024) and can be trivially increased by misclassifying specific images.

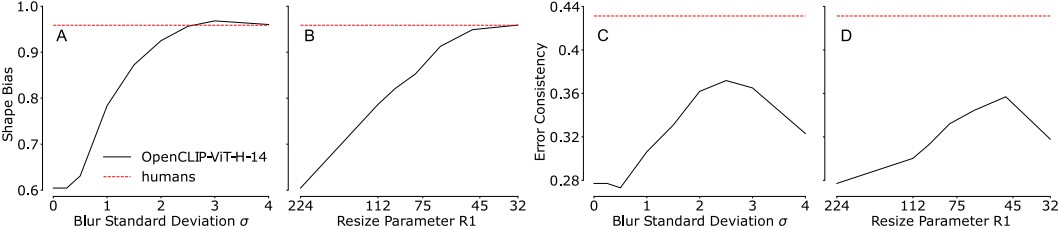

Figure 2: **Low-pass filtering test stimuli improves behavioral alignment.** Gaussian blurring [A+C] and resizing [B+D] both lead to higher shape bias (A+B) and error consistency (C+D). While the shape bias strictly increases under either transformation, the error consistency reaches a maximum "critical point" and declines afterwards.

Together, these findings offer an alternative explanation for the increased behavioral alignment of Imagen: Its lower input resolution alone suffices to explain its alignment. Jaini et al. (2023) go as far as re-training a ResNet-50 on ImageNet-1k with diffusion noise as an augmentation to obtain a shape bias of $0.78$, but a standard ResNet-101 with a prepended low-pass filter ($\sigma = 3.0$) achieves a shape bias of $0.8$ without any extra training.

## 4.2 LEARNING AN OPTIMAL FILTER INDEED YIELDS A LOW-PASS FILTER.

Given that low-pass filtering yielded considerable improvements, we next wonder what the *optimal* filter would look like. To address this question, we learn a filter in Fourier space for maximum error consistency using gradient descent (see Section 3.3 for details). The resulting filter is depicted in Figure 3, where we plot the optimal Fourier filter. Since we deliberately parameterized the filter in a way that allows for rotational asymmetries (i.e. different directions being affected differently) to find an upper bound on the best possible filter, we observe some artifacts that could be a consequence of potential overfitting as speculated in Section 3.3. However, the filter's spectrum is clearly concentrated in the low frequencies.

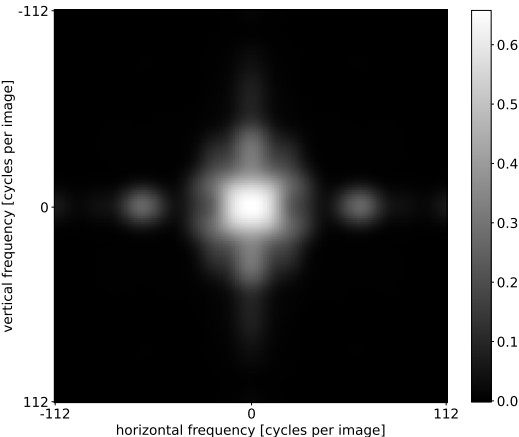

Figure 3: **The optimal filter is a low-pass filter.** The log-transformed, center-shifted amplitudes of the optimal filter, as a function of the horizontal and vertical frequencies (in cycles per image).

## 4.3 LOW-PASS FILTERING WORKS BECAUSE IT APPROXIMATES THE CSF.

The human visual system differs from DNNs in that images are not directly available to cortical neurons at full spatial resolution and bit-depth. Instead, the visual input is first impoverished by the imperfect optics of the eye, and later by the limited photoreceptor density on the retina and other neural factors. The degree of the reduction in perceived contrast depends on the spatial frequency of the input (Campbell & Robson, 1968), as well as the temporal frequency of its presentation

(Kelly, 1979). The overall reduction in perceived contrast is described by the contrast sensitivity function (CSF), which gives the inverse of threshold contrast as a function of spatial and temporal frequency of a stimulus. In figure Figure 4, we show that the spectrum of our best-performing low-pass filters approximates the CSF at 200 ms quite well. Interestingly, Li et al. (2022); Akbarinia et al. (2023) have found evidence for human-like CSFs in DNNs, but specifically CSFs matching the standard human CSF for unlimited viewing time. If networks already exhibit a CSF that matches the human CSF at long viewing times, an adaptation has to be implemented to account for the shorter presentation times—which low-pass filtering accomplishes. We thus interpret our results to mean that by prepending models with low-pass filters, we effectively match their input to that available to the visual cortex of human subjects.

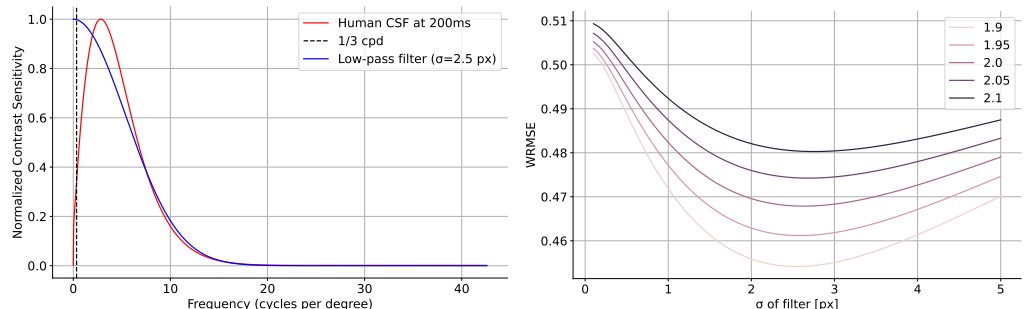

Figure 4: **Low-pass filters approximate the CSF well. Left:** Contrast sensitivity at a presentation time of 200 ms as a function of spatial frequency (red curve) is approximated well by the best Gaussian filter (blue curve). **Right:** The best-fitting Gaussian has a $\sigma$ of about $2.5px$, matching our empirical result. Evidently, this finding is robust to the exact choice of $\beta$.

Having established that Gaussian filters approximate the CSF well, we next test the effect of using the CSF as a filter directly. Indeed, this yields an error consistency of $0.365$, which beats Imagen and is statistically indistinguishable from the ideal Gaussian. We thus conclude that the improved error consistency between humans and DNNs is caused by providing DNNs with images whose spectra are filtered in a way that approximates how the human visual cortex receives its input: Diffracted by the optics of the eye and processed by the first few neural transformations, presumably in the retina and the lateral geniculate nucleus (LGN).

| model | error consist. ↑ | shape bias ↑ | OOD acc. ↑ |
|---|---|---|---|
| Humans (avg) | 0.43 | 0.96 | 0.72 |
| ViT-22B-384 | 0.26 | 0.87 | 0.80 |
| OpenCLIP ViT-H-14 | 0.28 | 0.60 | 0.78 |
| Imagen | 0.31 | 0.99 | 0.71 |
| OpenCLIP ViT-H-14 Resized (64x64) [ours] | 0.35 | 0.91 | 0.75 |
| OpenCLIP ViT-H-14 Blurred ($\sigma = 2.5$) [ours] | 0.37 | 0.96 | 0.72 |
| OpenCLIP ViT-H-14 Fourier-filtered [ours] | 0.38 | 0.95 | 0.73 |

Table 1: **Low-pass filtering improves error consistency and shape bias.**

### 4.4 LOW-PASS FILTERING WORKS ACROSS MODELS.

A natural next question is whether the blur and resize transformations only increase human-model error consistency for the OpenCLIP ViT-H-14 model, or whether this is a more general phenomenon which applies to other models as well. To answer this question, we test several models included in the `model-vs-human` package, as well as other OpenCLIP models, with prepended low-pass filters. We find that our approach generalizes almost perfectly: Shape bias and error consistency increase with low-pass filtering for almost all tested models. The effect is particularly strong for the OpenCLIP models, especially those with smaller patch sizes of 14 instead of 32 pixels (see Figure 5). The reason might be that a smaller patch-size biases a model towards high-frequency texture features, which are strongly affected by these transformations.

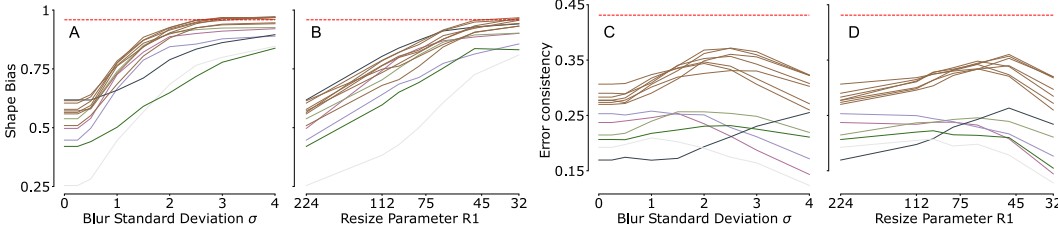

Figure 5: **Removing high-frequency information from test stimuli improves behavioral alignment for a wide range of models.** We measure shape bias and error consistency for ResNet; SWSL, BiT-M, ViT, Noisy Student, and a variety of OpenCLIP models. We find that in general, shape bias [A+B] increases with blur and resize strength, while error consistency [C+D] usually increases at first before dropping off again.

## 4.5 THE ACCURACY-CONSISTENCY-TRADEOFF

Because OOD accuracy and error consistency are at odds, the ceiling performance on model-vs-human is not a single point, but a curve of pareto-optimal solutions, which we call the pareto-frontier. The maximum possible error consistency to human observers is $0.674$, necessitating an OOD accuracy of $0.67$. As the OOD accuracy approaches $100\%$, error consistency approaches $0$. Between these two extremes, there is a smooth curve of pareto-optimal solutions, which we plot in Figure 6. Since these values are optimal and thus fixed, there is no uncertainty about them, so we plot no confidence intervals. We relegate detailed explanations of how we arrived at these values to Appendix A.3, where we also explain the counter-intuitive finding that the maximum EC extends beyond the average inter-human EC. We also find hints of this curve empirically in the models we investigate, see Figure 6 (right).

While our filtered models achieve performance values closest to the noise ceiling of inter-human error consistency, there is still plenty of room for improvement, especially in terms of improving OOD accuracy without sacrificing alignment to humans. The MvH benchmark is clearly not saturated, but we note that it suffers from some idiosyncrasies—for example, there are trivial ways of achieving high shape bias (Doshi et al., 2024), and error consistency is known to be noisy (Klein et al., 2025).

However, we still believe the trade-off between behavioral alignment and accuracy to be more fundamental: A machine that reaches super-human accuracy is bound to use features that humans do not use, which should generally result in a misalignment of their behavior. By carefully imposing limitations on the model (in our case, limiting the frequency content of its input via low-pass filtering), more human-like features may be extracted, albeit at the cost of accuracy.

## 4.6 LIMITATIONS

**Neural Similarity Measures.** A natural extension of this work would be to evaluate the effect of test-time low-pass filtering on neural similarity scores as well. Ideally, one would compare the neural predictivity afforded by our most human-like models to that of many other models, across different datasets of neural recordings. Such comparisons are made possible by the Brain-Score platform (Schrimpf et al., 2018). However, model inputs in Brain-Score are standardized to subtend eight degrees of visual angle. To achieve this, MvH images are padded with mean gray pixels, which amounts to a downscaling operation that conflicts with our test-time filtering. (Notably, as a consequence, error consistency results on Brain-Score do not match model-vs-human.) We would thus have to drastically change the preprocessing pipeline of Brain-Score to evaluate our approach, and then re-evaluate all other models for a fair comparison. Furthermore, the presentation times for different neural recordings might have been different, meaning that different low-pass filters would be optimal for different datasets. Hence, we leave an analysis of the effect of test-time low-pass filtering on neural alignment to future work.

**Longer Exposure Times.** Another logical question is how test-time low-pass filtering affects a model's EC to humans *at longer presentation times*, when the CSF looks more like a band-pass filter with a peak at 3 degrees of visual angle rather than a low-pass filter. However, evaluating

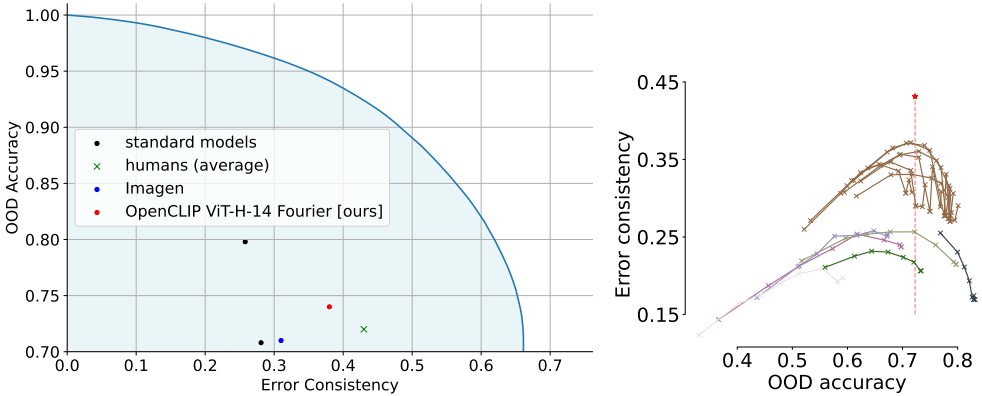

Figure 6: **Pareto Frontier of MvH solutions. Left:** We plot error consistency against OOD-accuracy, and delineate the region of optimal achievable benchmark performance. We plot the performance of various standard models as reference points, as well as Imagen (blue point) and average human performance (green x). Our CLIP ViT-H-14 with prepended low-pass filter achieves performance closest to humans and improves over Imagen on both axes. **Right:** Empirical confirmation of this result. Models from OpenCLIP tend to achieve the highest error consistency when their OOD accuracy coincides with that of humans (dashed red line).

EC is problematic at longer presentation times, for practical reasons. Human observers will make fewer mistakes on MvH, leading to the ceiling performance issues discussed in Klein et al. (2025). Furthermore, human error patterns themselves become more erratic, as random mistakes caused by attentional blips or motor noise dominate the errors. Another issue would be that at presentation times in excess of 200 ms, humans will make multiple saccades, moving the experiment away from the domain of perception into the domain of visual reasoning, presumably leading to different error patterns. We thus do not explore this avenue here.

## 5 CONCLUSION

In this work, we explained the high error consistency observed for the Imagen-model by its resizing operation rather than its generative objective. This resizing amounts to low-pass filtering of input images *at test-time*, in contrast to other works like Jang & Tong (2024) who achieved modest EC gains by low-pass filtering at train-time. We showed that by simply prepending discriminative classifiers with an appropriate low-pass filter, we can achieve even higher behavioral alignment, an effect which generalizes to the majority of tested models. We then demonstrated that a low-pass filter is indeed optimal by actively optimizing a filter for error consistency in Fourier space, resulting in a low-pass filter.

We offer a potential explanation for this effect: By approximating the human contrast sensitivity function (CSF) at short presentation times, an ideal low-pass filter approximates how the human visual cortex receives its input. Incoming light is diffracted by the optics of the eye and subjected to initial neural transformations, presumably in the retina and the lateral geniculate nucleus (LGN). Various attempts have been made to *train* DNNs on a diet of (initially) blurred images (Vogelsang et al., 2018; Jang & Tong, 2024; Lu et al., 2025), but our results suggest that if one matches a blurring filter to the human visual system, re-training of models might not be necessary to achieve improved behavioral alignment to humans. This intuitively makes sense, since training models on low-pass filtered images might render the model less susceptible to such corruptions, leading to changed error patterns. Using the contrast sensitivity function as a filter establishes a new state-of-the art error consistency on the model-vs-human benchmark. To contextualize this result and answer the question of how much room for improvement is left, we compute the frontier of pareto-optimal solutions to the benchmark, which was formerly unknown. The frontier reveals that further performance gains are possible, even though they would have to exceed the noise ceiling of the benchmark. Together, these results imply that generative objectives may in fact not be necessary to achieve higher alignment to human observers, with implications about the role of generative, "top-down" structures in the human visual system.

## 6 REPRODUCIBILITY STATEMENT

We are committed to ensuring the reproducibility of our work. All code required to reproduce the experiments is provided in the supplementary material. Detailed proofs of all theoretical results are included in the appendix. Experimental settings and hyperparameters are described in the main paper and supplementary sections.

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

# A APPENDIX

## A.1 ILLUSTRATIONS OF TRANSFORMATIONS AND FILTERS

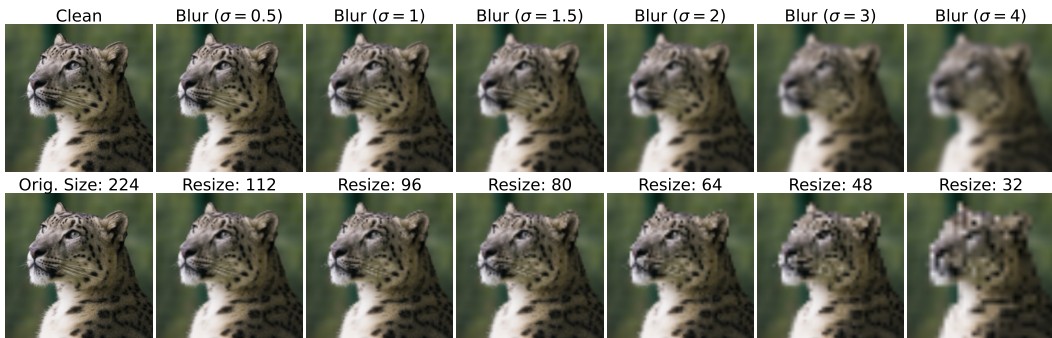

Figure 7: **Illustration of blurring and resizing filters.** Best viewed on screen. Evidently, the effects of low-pass filtering (implemented as convolution with a Gaussian kernel) and resizing with bi-cubic interpolation are very similar: Resizing is a form of low-pass filtering.

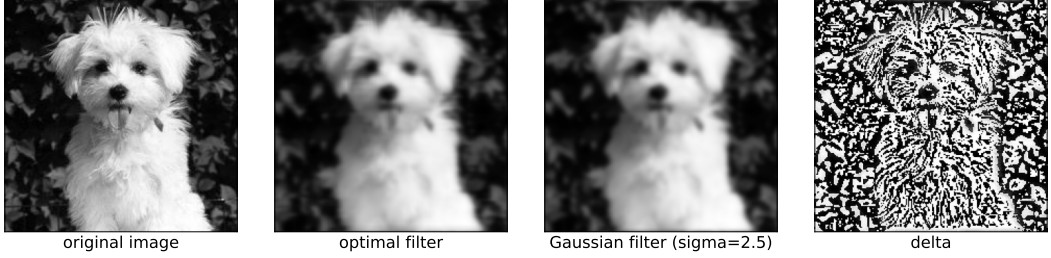

Figure 8: **Illustration of the learned optimal filter.** Best viewed on screen. We compare the effect of our learned optimal filter to that of the optimal Gaussian ($\sigma = 2.5$), which is visually indistinguishable, and plot the absolute delta between the resulting images to show that there is indeed a difference.

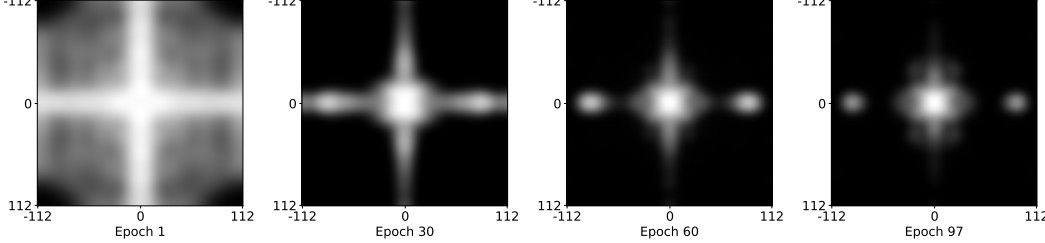

Figure 9: **Progression of filter training.** Best viewed on screen. We visualize the intermediate results of learning the optimal filter, analogous to Figure 3. The best filter is found at training epoch 97. Evidently, the diagonal frequencies decay quickly and equally in all directions, while the horizontal and vertical frequencies take longer, but eventually subside as well.

## A.2 MODELS OF HUMAN CONTRAST SENSITIVITY

The influence of the optics of the human eye on the light received at the retina is described by the modulation transfer function (MTF) (Campbell & Green, 1965; Williams et al., 1994). The MTF describes how well an optical system such as the eye transfers contrast from the object to the retinal image at different spatial frequencies, before any neural processing takes place. The losses caused by

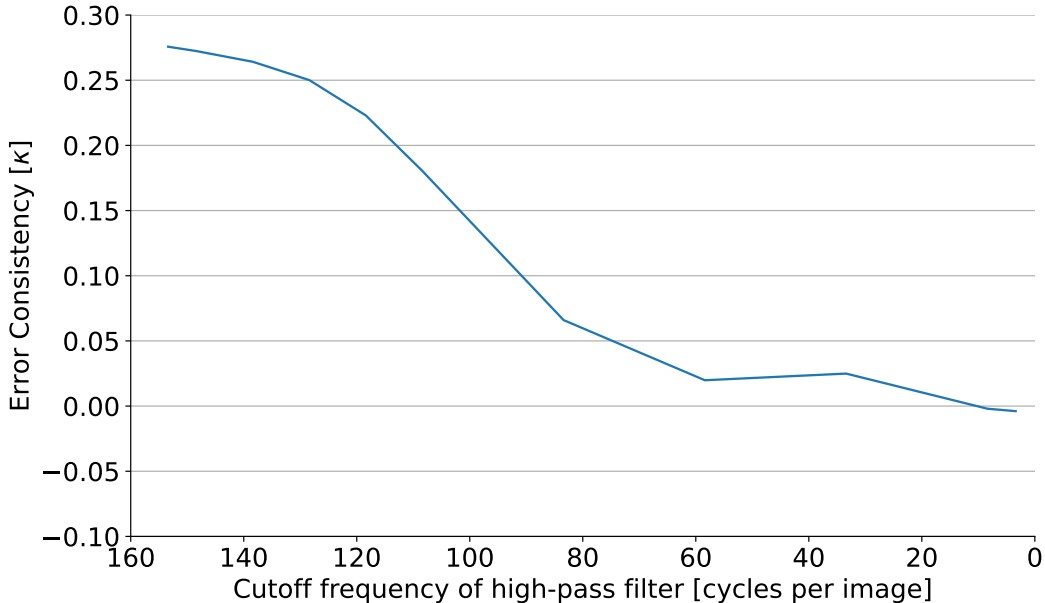

Figure 10: **Ablation: High pass filtering.** We test the effect of increasingly aggressive high-pass filters (i.e. decreasing cut-off frequency) on error consistency to humans. We observe quite drastic accuracy drops for these filters, resulting in low error consistency values. The highest frequency contained in the image is given by $\sqrt{2} * 112 = 158.39$, so we test values between this maximum value and 10 cycles per image.

low photoreceptor density on the retina and other downstream effects are incorporated in the contrast sensitivity function (CSF), but not the MTF. We compare the MTF to the Gaussian filter with optimal error consistency in Figure 11. While the fit may not seem great at first glance, note that the spectrum of natural images is typically $f^{-1}$, so we weight the frequencies by their contribution to overall power when evaluating goodness of fit.

We use the empirical estimate of the MTF obtained by Williams et al. (1994), who measured the MTF using interferometry. Their MTF models the eye as a diffraction-limited system, resulting in the formula

$$M(f_s, s_0) = D(f_s, s_0)(w_1 + w_2 e^{-as}) \tag{4}$$

where $f_s$ is the spatial frequency and $s_0$ is a constant depending on pupil size (we assume a $3\,\mathrm{mm}$ pupil size for $s_0 = 87.2$ cpd). The parameters $w_1$, $w_2$ and $a$ are obtained by fitting this curve to empirical data. $D(f_s, s_0)$ describes the modulation transfer of such a diffraction-limited system as

$$D(f_s, s_0) = \frac{2}{\pi} \left( \cos^{-1}\left(\frac{f_s}{s_0}\right) - \frac{f_s}{s_0} \sqrt{1 - \left(\frac{f_s}{s_0}\right)^2} \right). \tag{5}$$

For the CSF, we use the model by Kelly (1979), who model contrast sensitivity as a function of spatial frequency and retinal velocity (i.e. how fast the image moves across the retina):

$$G(f_s, v) = kvf_s^2 e^{\frac{-2f_s}{s_{max}}} \tag{6}$$

where $f_s$ is again the spatial frequency in cpd, $v$ is the retinal velocity of the stimulus in degrees per second, while $k = 6.1 + 7.3|log(\frac{v}{3})|^3$ and $s_{max} = \frac{45.9}{v+2}$ are scale parameters. From $G$ we can obtain an expression that gives the contrast sensitivity as a function of spatial and temporal frequency, $S(f_s, f_t)$, by noting that retinal velocity and temporal frequency are related: $v = f_t/f_s$. We refer

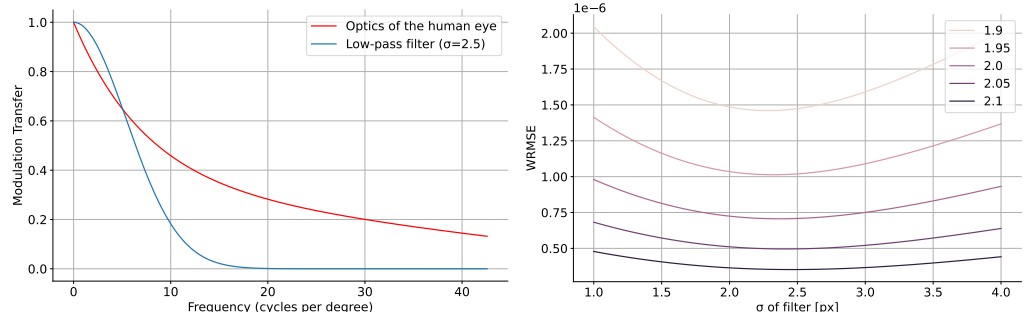

Figure 11: **Low-pass filters approximate the MTF relatively well for low frequencies. Right:** The spectrum of the MTF and our optimal Gaussian filter. **Left:** The goodness of fit between Gaussian filters of varying $\sigma$ and the MTF, calculated as a power-weighted RMSE. Both analogous to Figure 4.

the interested reader to the original papers for details and motivation of these models. (Note that at $f_s = 0$ the function is 0; we manually set the DC component of the filter to $1.0$ because we are concerned with contrast sensitivity, while the DC component gives the mean luminance.)

Matters are complicated by the fact that Kelly (1979) was concerned with stimuli of pure sinusoidal temporal frequency, that is, flickering or moving gratings. However, the stimuli in MvH are presented with hard temporal stimulus on- and offset, i.e. a boxcar function in time. The Fourier transformation of a boxcar function of duration $T$ is given by the $sinc$ function: All temporal frequencies $f_t$ contribute with coefficients given by $sinc(\pi f_t T)$, so the boxcar-CSF $S_{box}$ for spatial frequency $f_s$ and duration $T$ is given by

$$S_{box}(f_s, T) = \left[ \int S(f_s, f_t)^2 T^2 sinc^2(\pi f_t T)\, df_t \right]^{1/2}. \tag{7}$$

The appropriate CSF in the context of MvH is thus $S_{box}(f_s, 0.2)$.

A crucial step for using these formulas to construct the appropriate 2D filters is that the input scales need to be considered: Both MTF and CSF as formulated above expect frequencies in cycles per degree, while the DFT in our implementation is defined in terms of cycles per image. To take this into account, we specify a sampling rate of $\frac{3}{256}$ (because the images were presented at a size of $256 \times 256$ pixels, covering 3 degrees of the visual field of human observers in MvH) thus matching the inputs appropriately.

### A.3 Computing the Pareto Frontier of model-vs-human

For a detailed explanation of error consistency, we refer the interested reader to Klein et al. (2025), but we will summarize the relevant details here. The model-vs-human benchmark consists of 17 *corruptions* (e.g. inverted colors, eidolon noise, etc) with different 46 different *conditions* arising due to varying corruption strengths. Within every condition of model-vs-human, the error consistency is calculated via Cohen's $\kappa$ (Cohen, 1960) over binary vectors indicating whether a trial was solved correctly or not:

$$\kappa = \frac{p_{obs} - p_{exp}}{1 - p_{exp}}. \tag{8}$$

Here, $p_{obs}$ is the observed consistency, i.e. the number of trials on which both observers agreed by either both responding correctly, or by both responding incorrectly. $p_{exp}$ is the agreement expected by chance: Let $a_1$ and $a_2$ be the accuracies of the two classifiers, i.e. a model and a human subject. Then, $p_{exp} = a_1 \cdot a_2 + (1 - a_1) \cdot (1 - a_2)$ (i.e. $p_{exp}$ is the agreement expected by chance, assuming independent binomial observers).

A model's error consistency to humans within a condition is defined as its average (pairwise) error consistency to each of the human observers. Its overall error consistency is then calculated by

hierarchical averaging: First across the conditions of a corruption, then across corruptions. The inter-human error consistencies are calculated in the same way, with only a subtle difference: Each of the $n$ humans is only compared to the remaining $n-1$ humans. The green x in Figure 6 is the (hierarchical) average over these inter-human values. Hence, it is possible to achieve error consistency to humans exceeding the average inter-human error consistency. For an intuitive example, consider a human who only gave correct responses and another human who only gave incorrect responses. By responding correctly to half of all trials, a model would be more consistent to both humans than they are to each other. As the OOD accuracy of a model approaches $100\%$, its error consistency to humans necessarily approaches $0$, because the bounds on $\kappa$ get "squished" (compare Figure 2 from Klein et al. (2025)).

Our goal is to find the maximum attainable average $\kappa$ to the human observers. Let $T \in \{0,1\}^{n \times N}$ hold the $n$ fixed binary correctness sequences for this condition, with each observer being one row of $T$, and each image a column. Let $s \in \{0,1\}^N$ be the response vector that we get to optimize, so that $\kappa$ becomes maximal. The key insight[2] is that $\kappa$ can be expressed as

$$\kappa_j(s) = \frac{2(r_j - pq_j)}{p + q_j - 2pq_j} \tag{9}$$

where

- $p = \frac{1}{N} \sum_i s_i$ is the accuracy of the ideal model
- $q_j = \frac{1}{N} \sum_i T_{i,j}$ is the accuracy of human $j$
- $r_j = \frac{1}{N} \sum_i s_i T_{j,i}$ is the proportion of positive agreement between them.

We arrive at Eq. (9) via the following derivation:

$$\begin{aligned}
p_{obs} &= P(s = T_j) \\
&= P(s = 1, T_j = 1) + P(s = 0, T_j = 0) \\
&= r_j + P(s = 0, T_j = 0) \\
&= r_j + 1 - P(s = 1) - P(T_j = 1) + P(s = 1, T_j = 1) \\
&= 1 - p - q_j + r_j
\end{aligned}$$

Similarly, $p_{exp} = 1 - p - q_j + 2pq_j$. By using these expressions in Eq. (8), we obtain Eq. (9).

Conveniently, for a fixed $p = \frac{k}{N}$, the denominator $D_j(p) = p + q_j - 2pq_j = q_j + (1 - 2q_j)p$ is a constant with regard to the individual $s_i$. The numerator of Eq. (9) is thus a linear function of $s$ under the cardinality constraint $\sum_i s_i = k$. This linear objective has weights $w_i(p)$, given by

$$w_i(p) = \sum_{j=1}^{n} \frac{T_{j,i} - q_j}{D_j(p)}. \tag{10}$$

The optimal binary sequence for a fixed $k$ can therefore be obtained by setting $s_i = 1$ for the top $k$ items according to $w_i(p)$ and $s_i = 0$ for the others. The global optimum can then be obtained by simply sweeping $k$ from $0$ to $N$, obtaining the optimum $\kappa$ for each value of $k$, and selecting the overall best $\kappa$.

Since the model-vs-human benchmark consists of multiple such conditions, this procedure needs to be applied to each condition independently, and results need to be aggregated using the hierarchical averaging employed by the benchmark: First, one has to average over all conditions within one experiment, before then averaging over all experiments. This procedure yields the maximum possible error consistency and the sequence $s$ of ideal responses, from which the accuracy can be obtained by following the same hierarchical averaging logic. Thus, one has computed the lower right point on the pareto-frontier in Figure 6.

---

[2]The following derivation, while carefully verified by the second author, was obtained with heavy LLM support, as discussed in Appendix A.6.

The rest of the pareto-frontier can be constructed by iteratively expanding the frontier. Note that for every condition of the experiment, we have a list of tuples $(\kappa_i, p_i)$. From this list, we can remove all dominated elements $j$, for which there is another element $i$ so that $(\kappa_j \leq \kappa_j)$ and $(p_j \leq p_i)$, because these clearly cannot be part of a non-dominated solution. Next, we initialize the set of non-dominated solutions as $S_0 = \{0, 0\}$. Then, for each condition $i$:

1. Let $O_i$ be the set of non-dominated elements in that condition.
2. Expand $S_{i-1}$ by forming all possible combinations: $S_i = \{(\kappa, p) + (\kappa', p') : (\kappa, p) \in S_{i-1}, (\kappa', p') \in O_i\}$.
3. Prune $S_i$ by removing all dominated solutions.

To account for the fact that each condition contributes with a different weight to the overall average (because the number of conditions $C$ per experiment varies), we first weight each condition's elements with the appropriate factor $\frac{1}{C}$. The worst-case runtime of this algorithm is exponential, but at the small scale required for model-vs-human, we could calculate the full pareto-frontier on consumer hardware in less than ten minutes.

## A.4 Model Details

We present an overview of the models used in this work in Table 2.

| model | architecture |
| --- | --- |
| ResNet (He et al., 2016) | ResNet101 |
| SWSL (Yalniz et al., 2019) | ResNeXt-101 |
| BiT-M (Kolesnikov et al., 2020) | ResNet-101x1 |
| ViT (Dosovitskiy et al., 2020) | ViT-L-16 (IN1K and IN21K) |
| Noisy Student (Xie et al., 2020) | EfficientNet-L2 |
| OpenCLIP (Ilharco et al., 2021) | ViT-B-32 |
| | ViT-B-16 |
| | ViT-L-14 |
| | ViT-H-14 |
| | ViT-g-14 |
| | ViT-G-14 |
| | ConvNext-L |

Table 2: Architectures and the color coding used for each model type. We adopt these from Geirhos et al. (2021).

## A.5 Detailed Results

Here, we provide additional results of our analyses. In Figure 12, we show confidence intervals for the measured error consistency values of CLIP ViT-H-14 prepended with the optimal low-pass filter and our learned filter, as well as inter-human error consistency. Evidently, we almost close the gap to human observers.

## A.6 LLM Usage

The first authors of this submission use IDEs with built-in LLM support, so LLMs have been used to help with menial coding tasks. Beyond that, the authors have used chat-GPT5-Pro to aid with verifying code or high-level approaches to problems in ways that we deem uncontroversial, with one exception: To err on the side of caution, the authors take no credit for Eq. (9) and Eq. (10). While we have carefully verified that this solution is indeed correct, chat-GPT5-Pro derived this result autonomously when prompted to do so with a detailed problem description and definition of error consistency. The idea of searching for the pareto-frontier in the first place (which we deem the more important intellectual contribution than the algebraic manipulation) is fully our own. LLM tools have also been used only very sparingly in writing, not more than one would use a thesaurus or dictionary.

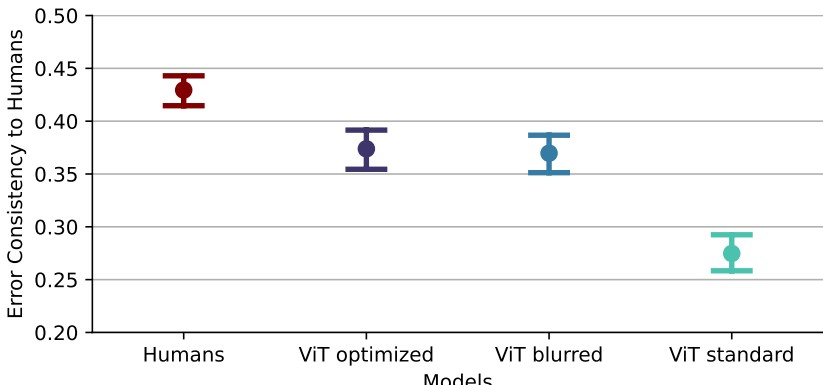

Figure 12: **Error Consistency to humans.** We plot the error consistency to human observers as measured by the model-vs-human benchmark for a vanilla OpenCLIP ViT-H-14 as well as the same model with a prepended low-pass filter ($\sigma = 2.5$) and a filter learned in Fourier space. Confidence intervals are obtained via bootstrapping as described in Klein et al. (2025). See Figure 14 for a breakdown by corruption.

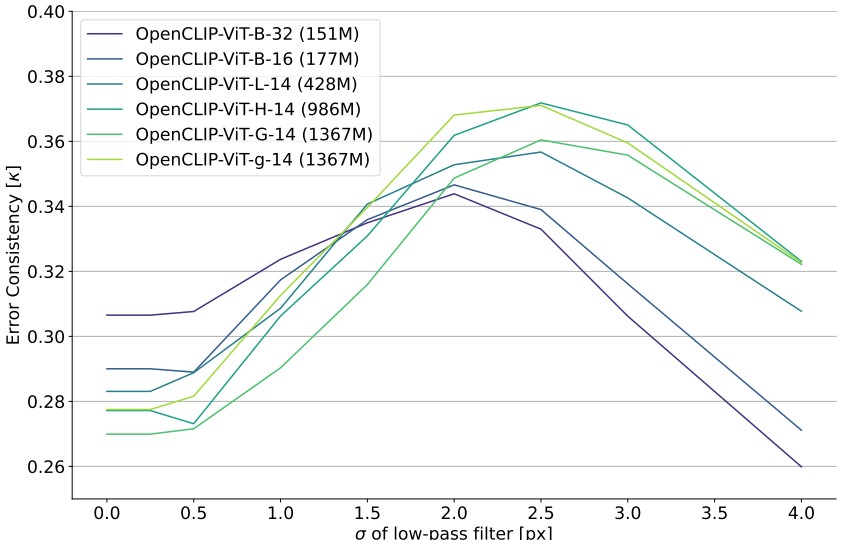

Figure 13: **Optimal $\sigma$ as a function of model size.** We plot the error consistency achieved for a certain filter $\sigma$, for each OpenCLIP ViT size in terms of parameter count. Evidently, the optimal $\sigma$ is fairly robust to the model size, and tends to increase a bit for larger models / smaller patch sizes.

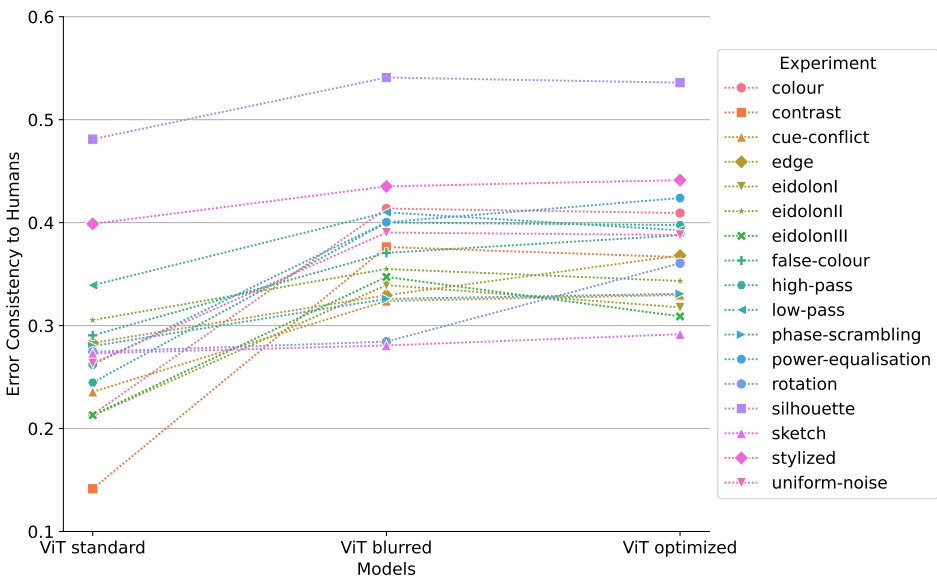

Figure 14: **Breakdown of EC gains by corruption.** We plot the error consistency to human participants that every model achieves, broken down by corruption. Evidently, the EC gains are consistently achieved across all corruptions.

