# OpenReview forum: "Low-Pass Filtering Improves Behavioral Alignment of Vision Models"
_ICLR.cc/2026/Conference — ICLR 2026 Poster_

### Official Review · Reviewer_V1fP · 2025-10-29

**Soundness:** 2
**Presentation:** 3
**Contribution:** 3
**Rating:** 6
**Confidence:** 4

**Summary:**

This paper demonstrates that applying low-pass filters (gaussian blur and resampling) during test time improves model alignment with human perception in the context of the model-vs-human (Geirhos et al., 2021) benchmark framework. The authors claim that this result is further explained by established models of human visual sensitivity in the context of fast presentation times. To support this framework, an optimal filter that maximizes alignment is also learned and was shown to be, qualitatively, a low-pass filter.

**Strengths:**

The work addresses a topical problem of human-model alignment, specifically in an overlooked area of alignment, error consistency.

It is a particularly simple and elegant hypothesis and result that a simple low-pass filter can improve error consistency, and as far as I know, this is a novel result.

The manuscript is well written and communicated; the authors do a good job at leading the reader from the motivation, background to their results and conclusions

Prior work on human visual processing well covered and incorporated into the work; the authors cite well known, tested, and accepted results from the human vision literature.

While the experiment of blurring images is generally simple, the authors explore many different methods and levels of blur, and the results demonstrating a peak of error consistency for a given blur is interesting.

**Weaknesses:**

The majority of the weaknesses of the paper comes from the connection to human vision. While well cited, there are many shaky logical steps to support the claim that this improvement in error consistency with human data is because of a match in a low-pass-filtered image to the human percept of an image when viewed for only 200ms. Generally, I feel the result of improvement in alignment in itself with just a simple blur is interesting, and the shaky claims that this blur amount is a match to the effective blur of human vision in the context of a short presentation time detract from this.

The largest example of this is a major discrepancy between Kelly’s work on contrast sensitivity to spatiotemporal frequency and limited presentation time of images. Specifically, in Kelly’s work, two types of stimuli were presented, a flickering grating of the form
$$
f_S(x, t) =\cos\alpha x \cos \omega t
$$
And a moving grating of the form
$$
f_T(x, t) = cos \alpha (x - v t)
$$
In both cases, the time dependencies are sinusoidal with a precise frequency whereas in the case of limited presentation of stimuli, the time dependency is a boxcar function (with a Fourier transform of a sinc function which has a peak, in fact, at f = 0). It is erroneous to take the inverse presentation time to be the frequency.
While Kelly's CSF might still be the best model available, the authors should at the very least acknowledge this discrepancy.

**Questions:**

The authors do not discuss how this result compares with other test-time pre-processing methods. Does previous work exist? What about a set of experiments that benchmarks various corruptions used in OOD-accuracy experiments from MvH (e.g. uniform noise, low contrast, etc.)

I recommend a key control experiment which would be to apply varying levels of high pass filtering as a test-time pre-processing manipulation.

While a human psychophysics experiment may be difficult to do during the rebuttal period, testing a subset of the images in humans with longer presentation time with and without blur would be the key experiment to determine if it is indeed the magno stream behind the human baseline.

Are the authors referring to the magno/parvo streams when they discuss the relationship to human perception, and claiming that the magno stream with it's low spatial frequency sensitivity is behind this low-pass-filtered effect described?

Minor Points:
A “neural transfer function” for short presentation times is mentioned in line 069 without prior introduction. Is this essentially the CSF?

Equation 1 has a $H_\theta$ that should probably be $G_\theta$, the learned filter

Equation 3 should have “df” to end both integrals on the numerator and denominator rather than being after the fraction.

On figure 3, there should be axis labels and the standard deviation of the learned filter (either in Fourier or real space) should be reported and compared against the gaussian blur filters. Furthermore, it could be helpful to show images filtered by this filter.

---

> ### Author Response · Authors · 2025-11-21
>
> Dear Reviewer,
>
> Thank you for your constructive and insightful review. We were delighted to read that you found our hypothesis **"simple and elegant"** and the manuscript **"well-written"**. We also greatly appreciate your expertise on the relevant vision science literature, and agree with your point that treating the inverse frequency as presentation time is an oversimplification due to the hard stimulus onset / offset employed by the model-vs-human benchmark. We have reworked parts of our manuscript to account for this, obtaining slightly better results than before; see below. Thank you for your helpful comments, we believe that they have greatly improved our submission.
>
> *“It is erroneous to take the inverse presentation time to be the frequency. While Kelly's CSF might still be the best model available, the authors should at the very least acknowledge this discrepancy.”*
>
> Thank you for this fantastic feedback. You are absolutely correct, this approximation was crude, and we should have pointed this out. Following your reasoning, we have now computed the proper CSF for a boxcar stimulus, by weighting the temporal components of Kelly’s spatiotemporal CSF according to the sinc-function, see Appendix A.2 of the newest iteration. The difference in CSFs is a bit larger than we had expected; the more accurate CSF peaks at slightly higher frequencies. We have recomputed figure 4 and now find an even better match between the $\sigma$ that leads to highest EC and the CSF: The optimal $\sigma$ predicted by the CSF is now indeed 2.5px, which matches what we find in practice. We are in the process of testing this CSF as a filter, to see whether it will lead to higher empirical EC to humans when prepended to a ViT-H-14 model.
>
> *“Does previous work [on other test-time preprocessing methods] exist?”*
>
> Thank you for raising this question. As far as we are aware, no prior work has explicitly investigated the influence of test-time preprocessing on error consistency, or behavioral alignment in general. We know of multiple works that propose train-time preprocessing (i.e. data augmentation), some of which achieve slightly higher error consistency. In a sense, the closest to our approach is [2], who prepend CNNs with a bank of Gabor patches that are not learned, but modelled after “empirically observed distributions of preferred orientation, peak spatial frequency, and size/shape of receptive fields” in primates. Since the Gabor patches are static, one could think of this as a preprocessing of images, but it again is applied at both train- and test-time.
>
> *“I recommend a key control experiment which would be to apply varying levels of high pass filtering as a test-time pre-processing manipulation.”*
>
> Thank you for this very sensible suggestion. We had tested an accuracy-matched high-pass filter for an ablation, to make sure that the gains in error consistency are not simply a consequence of a better match of model accuracy to human accuracy (because the theoretical upper bound on error consistency depends on classifier accuracy alignment, see [1]). Following your suggestion, we will conduct a more fine-grained analysis. Based on the earlier ablation, we expect high-pass filtering to lead to some improvement of error consistency, but to not exceed gains obtained from low-pass filtering.
>
> *“Is [the neural transfer function mentioned in line 069 without prior introduction] essentially the CSF?”*
>
> Yes, thank you for the question. We have rephrased this sentence to improve clarity. We meant to say that at the short presentation times (of 200ms) employed by the MvH benchmark, both the MTF and the CSF implement low-pass filters. We agree that the wording was confusing, we now write “Imagen's low-pass filtering might be the true reason for its increased human-like behavior, since both the optics of the human eye and early neural processing stages act as low-pass filters, at least at the short presentation times used in the MvH benchmark.”
>
> *“Equation 1 has a $H_{\theta}$ that should probably be $G_{\theta}$, the learned filter.”*
>
> Thank you for pointing out this mistake. You are correct, this should have been $G_{\theta}$, which we have now fixed. Our original draft made a distinction between the not-yet-smoothed raw filter, $H_{\theta}$, and the final filter $G_{\theta}$. We forgot to explain this in the text, hence the confusion. However, we have now decided that this distinction is needlessly confusing, and have changed the notation accordingly.
>
> [1] Klein, T., Meyen, S., Brendel, W., Wichmann, F. A., & Meding, K. (2025).Quantifying Uncertainty in Error Consistency: Towards Reliable Behavioral Comparison of Classifiers. In NeurIPS 39.
>
> [2] Dapello, J., Marques, T., Schrimpf, M., Geiger, F., Cox, D., & DiCarlo, J. J. (2020). Simulating a primary visual cortex at the front of CNNs improves robustness to image perturbations. Advances in Neural Information Processing Systems, 33, 13073-13087.

---

> > ### Author Response · Authors · 2025-11-21
> >
> > *“Equation 3 should have “df” to end both integrals on the numerator and denominator rather than being after the fraction.”*
> >
> > Thank you for this suggestion, we agree and have made the change.
> >
> > *“On figure 3, there should be axis labels and the standard deviation of the learned filter [...] should be reported [...]. Furthermore, it could be helpful to show images filtered by this filter.”*
> >
> > Thank you for these great suggestions, we will implement them.
> >
> > *“Are the authors referring to the magno/parvo streams when they discuss the relationship to human perception, and claiming that the magno stream with its low spatial frequency sensitivity is behind this low-pass-filtered effect described?”*
> >
> > Thank you for the question. We are fairly agnostic with respect to the actual neural circuitry that gives rise to the characteristic CSF. Our central contribution is mainly the observation that low-pass filtering images at test time leads to increased alignment on the MvH benchmark, and that this effect suffices to explain the high performance of generative models like Imagen. We then noticed that the optimal filter matches the CSF quite well at exactly the presentation time which was used in the MvH benchmark. It thus seems reasonable to assume that an appropriate modulation of the available signal (in a sense, aligning it with the signal available to humans) would improve behavioral alignment in terms of which images are difficult to recognize. As reviewer tqZJ pointed out to us, there is even existing work showing that various DNN-based models of vision exhibit similar CSFs as humans at unlimited viewing time (that is, linear decoders of visual contrast trained on frozen networks exhibit a CSF), lending further support to the idea that a modulation of the available frequency spectrum might increase alignment.
> >
> > *“While a human psychophysics experiment may be difficult to do during the rebuttal period, testing a subset of the images in humans with longer presentation time with and without blur would be the key experiment.”*
> >
> > Thank you for the insightful suggestion. We agree that in principle, testing humans at longer presentation times would be a good idea. However, there are practical considerations that prevent us from conducting this experiment: Humans will make fewer mistakes at longer presentation times, so we would run into the ceiling performance issues discussed in [1]. Furthermore, human error patterns themselves become more erratic, as random mistakes caused by attentional blips or motor noise dominate the errors (we have observed both effects in yet unpublished psychophysical work). Another issue would be that at presentation times in excess of 200ms, humans will make multiple saccades, moving the experiment away from the domain of perception into the domain of visual reasoning, presumably leading to different error patterns.
> >
> > [1] Klein, T., Meyen, S., Brendel, W., Wichmann, F. A., & Meding, K. (2025). Quantifying Uncertainty in Error Consistency: Towards Reliable Behavioral Comparison of Classifiers. In The Thirty-ninth Annual Conference on Neural Information Processing Systems.
> >
> > [2] Dapello, J., Marques, T., Schrimpf, M., Geiger, F., Cox, D., & DiCarlo, J. J. (2020). Simulating a primary visual cortex at the front of CNNs improves robustness to image perturbations. Advances in Neural Information Processing Systems, 33, 13073-13087.

---

> > > ### Comment · Reviewer_V1fP · 2025-11-25
> > >
> > > Thank you for addressing all these points.
> > >
> > > The authors have engaged, and have addressed all the points in the rebuttal, however, these resolutions are contingent on the corresponding revisions being clearly incorporated into the updated manuscript. At present, I do not see the revised version, so I cannot verify whether the clarifications, corrections, and additional analyses described in the response have in fact been implemented. My assessment therefore assumes that the promised changes—particularly those involving the corrected CSF treatment, revised equations, improved figure presentation, and clarified framing around human-vision interpretations are reflected in the updated submission. If these revisions are indeed present in the revised version and accurately described, they would resolve the majority of my concerns.

---

### Official Review · Reviewer_tqZJ · 2025-10-30

**Soundness:** 4
**Presentation:** 4
**Contribution:** 4
**Rating:** 8
**Confidence:** 5

**Summary:**

The authors investigate how simple down-sampling or low-pass filtering operations increase the alignment between *artificial vision models* (either discriminative, bottom-up, or generative, top-down) and *human vision* in terms of error consistency, shape bias, and out-of-distribution accuracy. They find that low-pass filtering similar to the human Contrast Sensitivity Function is optimal to maximize error consistency with humans and to get similar shape-bias in image recognition.  This finding is important in the discussion about the appropriate strategy (discriminative or generative) for object recognition both in humans and machines.

**Strengths:**

The (apparently) simple experiments proposed here, removing high-frequency information either by down-sampling or by low-pass filtering the input images before feeding the models, actually addresses a *key discussion* in machine vision: is it better a bottom-up approach to analyze the images and then discriminate between classes? (as done in conventional discriminative classifiers), or is it better a top-down approach where one checks if the input is compatible to generated examples form a given class? (as done in the "generative" classifiers as the ones defined by Jaini et al. 23, ICLR 24). This question is also relevant in understanding human vision where both strategies are certainly applied (though it is not clear how).

The results presented here constitute an excellent counter-example that allows to tone-down the conjectures about the benefits of the generative approach exposed in the ICLR 24 spotlight paper of Jaini et al. 23. While Jaini et al. suggest that their system may be "more human" (in shape-bias and error consistency) because of the generative approach, this work shows that this similarity may be just due to the fact that the considered models work with down-sampled images, thus effectively giving the model a signal which passed through a bottleneck similar to the one happening at the human LGN. As a result, as the signal content given to the model approaches the content arriving to the human visual cortex, it is reasonable that (1) models working with such filtered inputs have a generally "more human" behavior, and (2) those models focus on shape rather than texture because the texture has been largely attenuated by the CSF-like filter.

This counter-example reasoning is nice, results confirm that the simple low-pass operation alone does the alignment job for discriminative models, and show that the optimal filter resembles the human filter.

Finally, the scientific question-explanation logic of the work is very well posed, which (I think) is unusual in certain ICLR papers: the authors first state an important question in a compelling way and propose a simple hypothesis as an alternative explanation. Then, they experimentally show that this alternative may be true, thus questioning recent work, and motivate further research.

**Weaknesses:**

Weaknesses in this work are minor, mainly limited to (a) some notation issues, (b) mention to works that show the emergence of human-like Contrast Sensitivity in artificial nets, and (c) clarification of the discussion between using the low-pass in training or test time.

(a) Notation in Eqs. 1-3 can be more clear. I eleborate in the "questions" box below.

(b) In the Related Work section the authors mention the work of Subramanian et al. 23 on the frequency response of ANNs. Other works (e.g. Li et al. J.Vision 2022 and Akbarinia et al. Neural Nets. 23) specifically adress the issue of the emergence of Contrast Sensitivities in ANNs which may be similar or different(!) to the CSF of humans depending on the task and on the architecture indicating a non-trivial interplay between these issues.

(c) Finally, the discussion about whether the low-pass filtering is (or is not) required in the training of the models to increase the alignment with humans is not clear (to me). The presented experiments certainly show that using it in test time increases the alignment with humans, but it is not clear (to me) the behavior one could get with training based on low-pass filtered images (as is the case in infants with blurred vision and with the built-in CSF-like LGN bottleneck). May be this issue should be further clarified.

**Questions:**

(a) Regarding the notation question, consider that \sigma is used for different things: the blur extent, the soft-max function, and the noise used to initialize the all-pass filter to be optimized. Moreover, in Eq.1, I understand that b_{\sigma} is (or should be?) the filter, G_{\theta}, -as in Eq.3-, no?. Additionally, the parameters of b_{\sigma} are H_{\theta}?... is this H an entropy? or is it a generic way to call the parameters? (H is certainly a cross-entropy in Eq.2, but H in Eq.1 is confusing -to me-). I get the ideas and they are fine, but notation is confusing (to me).

(b) Regarding related work, Li et al. 22 ( https://jov.arvojournals.org/article.aspx?articleid=2778843 ) and Akbarinia et al. 23 ( https://doi.org/10.1016/j.neunet.2023.04.032 ) show that this kind of bottleneck can also emerge when solving certain tasks implemented by certain (simple) architectures, but does not emerge in other (too deep) architectures.

(c) Please clarify the discussion about whether the low-pass filtering is (or is not) required in the training of the models to increase the alignment with humans.

(d) Incidentally (but this is a criticism/question to Jaini at al. 23 not to this work ;-) I think that the increased shape-bias found in their generative classification may not be surprising because the noise added to launch the inverse in the difussion process in Jaini et al. destroys the texture in the image, so naturally, the classification (finally done in the spatial domain) has to be more driven by shape. Could the authors elaborate on this?

---

> ### Author Response · Authors · 2025-11-21
>
> Dear Reviewer,
>
> Thank you for taking the time to review our work. We are happy to read that you think our work **"addresses a key discussion**, and find that the **"logic of the work is very well-posed"**. We have addressed the minor weaknesses you pointed out, by adjusting our notation and incorporating the related work you kindly pointed out to us. These works are indeed highly relevant and match our findings well: Both works find that DNNs, especially if trained on low-level visual tasks, exhibit CSFs that match the human CSF at unlimited viewing time, which looks quite different from the CSF at 200ms. It makes sense that if networks have a “natural” CSF that matches the human CSF at long viewing times, an adaptation has to be implemented to account for the shorter presentation times – which low-pass filtering accomplishes. Thank you very much for pointing out this highly relevant related work!
>
> *“[The] notation [in Section 3.3] is confusing”*
>
> Thank you for pointing out the overloading of notation, especially of $\sigma$. We agree that this was a bit unfortunate, since $\sigma$ is standard notation for both blur strength and the softmax function. We have now changed the respective section to denote the blur strength as $\gamma$ instead. We also sincerely apologize for the confusion about equation (1), where we accidentally used notation without explaining it: We implement the smoothness constraint on the filter $G_{\theta}$ by convolving the filter itself with a Gaussian. $H_{\theta}$ was supposed to be a “proto-filter”, i.e. the raw parameters of the filter before this blurring is applied, so that $G_{\theta} = b_{\sigma} * H_{\theta}$. We have now decided against making this distinction for didactic reasons, and changed equation (1) to simply refer to $G_{\theta}$, as you correctly thought it should. Thank you very much for reporting this issue; fixing it has clearly improved our manuscript.
>
> *“Please clarify the discussion about whether the low-pass filtering is (or is not) required in the training of the models.”*
>
> Thank you for the question, we have refined our discussion to better address this point. Our investigation is indeed only concerned with applying low-pass filtering at test time – during training, low-pass filtering is not required. Reviewer 5YdF had a similar question: While other works have explored incorporating low-pass filtering into the training process, doing so only has a small effect on alignment to humans in terms of error consistency, presumably because it renders models more robust to certain inputs. But to achieve high error consistency to humans, models need to make mistakes on the same images as humans, so increased robustness is not necessarily desirable.
>
> *“The diffusion process in Jaini et al. destroys the texture in the image, so naturally, the classification (finally done in the spatial domain) has to be more driven by shape. Could the authors elaborate on this?”*
>
> Thank you for the interesting thought. We tend to agree that the type of distortion applied to input images in the work by Jaini et al. introduces biases about which features remain intact and will thus drive the classification decision. One could indeed speculate that destroying the texture is the reason for increased shape bias, however, in principle, the diffusion process could repair the texture information, thus maintaining a texture bias. Investigating the effect of the initial corruptions on shape bias and error consistency could be an interesting project in itself.

---

### Official Review · Reviewer_mKdF · 2025-10-31

**Soundness:** 3
**Presentation:** 4
**Contribution:** 3
**Rating:** 8
**Confidence:** 3

**Summary:**

The paper claims that recent improvements in human–model agreement for vision systems mostly come from an overlooked preprocessing step that removes high-frequency detail. The authors show that applying a simple low-pass filter to images at test time makes standard discriminative models behave more like humans on a benchmark of human vs model errors. They confirm this in three ways. First, simple blur or downsampling increases human-like behavior without retraining. Second, a learned frequency filter that is optimized for alignment ends up looking like a low-pass filter. Third, a filter shaped by the human contrast sensitivity function produces similar gains, which links the effect to known properties of early vision. The paper also maps the trade-off between accuracy and alignment to clarify what levels of agreement are realistically achievable.

**Strengths:**

- The writing is clear, the narrative is well structured, figures effectively support the claims.
- The paper offers a simple, general, physiologically grounded intervention with immediate practical impact: prepend a low-pass filter at test time to improve human alignment without retraining.
- Reinterprets prior SOTA claims by pinning gains to an overlooked preprocessing step. The Pareto-frontier view of MvH is insightful.
- Broad cross-model evaluation; consistent trends; convergence of three lines of evidence

**Weaknesses:**

- Potential overfitting. The learned Fourier filter is optimized on MvH without a held-out set. But it makes sense in the context.
- l.290: Please redefine EC ans SB because it requires coming back to previous section to find their meaning.

**Questions:**

- Do the gains persist on other human datasets or tasks (e.g., free-viewing, longer exposures)?
- If you re-fit the Fourier filter on a subset of MvH and evaluate on held-out conditions, does it still outperform simple Gaussian/CSF filters?


In general, I enjoyed reading the paper, which was well written and interesting. I would strongly encourage acceptance once my minor comments have been answered.

---

> ### Author Response · Authors · 2025-11-21
>
> Dear Reviewer,
>
> Thank you for providing such valuable feedback on our work. We were delighted to read that you found our manuscript **“well structured”** and our results on the Pareto frontier **“insightful”**.
>
> *“[There is p]otential overfitting.”*
>
> Thank you for raising this point. We agree that training the model on all MvH data is suboptimal; we opted for this approach only because of the limited amount of data relative to the number of parameters. We agree that it is only acceptable in our context of finding the optimal filter. An alternative approach would be to assume radial symmetry of the filter, which would further reduce the number of learnable parameters. However, even if we trained such a filter on 80% of the MvH data, and evaluated on the remaining 20%, we would run into a problem: The error consistency metric is very trial-inefficient, so values obtained on such a test split would be very unstable. We would face huge confidence intervals on such measurements, see figure 3 from [3]. Furthermore, the assumption of radial symmetry is quite strong and would drastically limit the analysis, because it would limit the search space to band-pass filters.
>
> *“Please redefine EC and SB”*
>
> Thank you for this constructive and specific feedback, we have rephrased the respective session and agree that it was indeed beneficial to remind the reader what EC and SB are.
>
> *“Do the gains persist on other human datasets or tasks (e.g., free-viewing, longer exposures)?”*
>
> Thank you for the excellent question. In principle, we could check whether the alignment gains afforded by prepended low-pass filters persist on other behavioral or neural datasets, such as Brain-Score [1] or Odd-One-Out Accuracy [2]. At different presentation times, we would expect that filters of a different $\sigma$ work best, because the precise shape of the CSF changes. Unfortunately, as also discussed in our response to reviewer 5YdF, evaluating on Brain-Score was not possible due to conflicting preprocessing and varying presentation times: We would have to use a different filter for the various datasets aggregated in Brain-Score, and Brain-Score already applies a filter, which we would have to remove, so comparisons to other models would be unfair. We would effectively have to redefine the Brian-Score evaluation pipeline.
> The data in [2] was collected with unspecified presentation time and covers multiple saccades, so it is again unclear which $\sigma$ should be used. Furthermore, odd-one-out decisions are presumably driven not only by low-level perceptual cues, but from higher-level semantic considerations. Therefore, it seems unlikely that low-pass filtering would have the same effects. We have now dedicated a section of the manuscript to a discussion of this issue.
>
> [1] Schrimpf, M., Kubilius, J., Hong, H., Majaj, N. J., Rajalingham, R., Issa, E. B., ... & DiCarlo, J. J. (2018). Brain-score: Which artificial neural network for object recognition is most brain-like?. BioRxiv, 407007.
>
> [2] Muttenthaler, L., Dippel, J., Linhardt, L., Vandermeulen, R. A., & Kornblith, S. Human alignment of neural network representations. In The Eleventh International Conference on Learning Representations.
>
> [3] Klein, T., Meyen, S., Brendel, W., Wichmann, F. A., & Meding, K. (2025). Quantifying Uncertainty in Error Consistency: Towards Reliable Behavioral Comparison of Classifiers. In The Thirty-ninth Annual Conference on Neural Information Processing Systems.

---

> > ### Comment · Reviewer_mKdF · 2025-11-26
> >
> > Thank you for the answers to my comments. I appreciate the clarifications provided on the potential overfitting issue, the limitations of evaluating on other human datasets, and the updates regarding the EC and SB terminology.
> >
> > As stated in my initial review, my comments were minor, and I remain in favor of acceptance once they are fully addressed in the manuscript. However, although the authors mention that revisions have been made, these changes are not reflected in the current manuscript version available on OpenReview.
> >
> > I would be happy to provide a definitive recommendation once a revised manuscript is uploaded, ideally with highlighted or clearly marked changes, so that the incorporation of the feedback can be properly assessed.

---

### Official Review · Reviewer_5YdF · 2025-11-03

**Soundness:** 3
**Presentation:** 3
**Contribution:** 3
**Rating:** 6
**Confidence:** 4

**Summary:**

This paper revisits the source of human-like behavior in modern vision models. Building on prior work suggesting that generative objectives (e.g., Imagen) yield superior behavioral alignment with humans, the authors demonstrate instead that the key factor is low-pass filtering, specifically, the effective downsampling to 64×64 resolution used in Imagen. Through extensive experiments across multiple architectures (ResNet, BiT-M, EfficientNet, OpenCLIP ViTs), they show that removing high-frequency content at test time (via Gaussian blur, resizing, or learned Fourier filtering) consistently: a) Raises shape bias (SB) toward human levels; b) Boosts error consistency (EC) beyond Imagen’s previous record (κ = 0.37 vs. 0.31); c) Only modestly reduces OOD accuracy (≈3–6 p.p.), revealing a clear accuracy–consistency trade-off curve.

The authors also derive a learned Fourier filter that optimizes EC directly, finding it converges to a low-pass profile nearly identical to the analytical Gaussian filter. Finally, they analyze the model-vs-human benchmark’s Pareto frontier, establishing the theoretical ceiling for behavioral alignment and showing that current models are still below this limit.

**Strengths:**

The paper is well-organized and well-written, with smooth narrative flow from hypothesis to methods to results. I think the paper provides a simple yet powerful reinterpretation of why generative vision models appear more human-like,  reframing a potentially deep theoretical question (generative vs. discriminative objectives) as a signal-processing issue (frequency content). As well, The authors test across numerous architectures, including large-scale CLIP models, and show strong generalization of the low-pass effect. I think they do a great job by providing a theoretical  link to the contrast sensitivity function (CSF) of human vision provides a compelling neurophysiological explanation, integrating psychophysical and computational perspectives.

**Weaknesses:**

The psychophysical alignment argument can  be strengthen  by testing on neural benchmarks (e.g., Brain-Score, Algonauts) to verify that low-pass filtering also aligns representational geometry with biological vision.

Also a control, may include testing under longer exposure times or more naturalistic conditions which could  clarify whether the low-pass benefit is tied to the MvH’s 200 ms presentation constraint or generalizes across temporal regimes.

While the learned Fourier filter converges to a low-pass profile, visualizing intermediate optimization stages could reveal whether specific spatial frequencies carry distinctive importance for behavioral alignment.

**Questions:**

Have you tested whether low-pass filtering also improves alignment on neural datasets (e.g., Brain-Score or fMRI RDMs)?

How sensitive is the optimal σ to model scale and patch size?

Could low-pass filtering be integrated during training to recover the lost OOD accuracy?

Is the Pareto frontier analysis robust to the noisy EC estimates reported by Klein et al. (2025)?

---

> ### Author Response · Authors · 2025-11-21
>
> Dear Reviewer,
>
> Thank you for taking the time to review our paper. We are delighted that you found our paper to be **“well written and structured”**, and appreciated the **“theoretical link to the human CSF”**. Based on your feedback, we have amended the manuscript to elaborate further on evaluating neural alignment and longer exposure times, see below.
>
> *“Have you tested whether low-pass filtering also improves alignment on neural datasets?”*
>
> Thank you for this suggestion. We had indeed considered this. However, Brain-Score enforces a unified preprocessing of images, standardizing all inputs to subtend 8 degrees of visual angle. To evaluate EC, inputs are thus padded with grey pixels, effectively introducing a downscaling operation that conflicts with the application of our filter. (Notably, as a consequence, EC results on Brain-Score do not match MvH.) Furthermore, the presentation times between different data sources of Brain-Score differ, so the filter that best matches each experiment would have to be chosen first.
> Thus, to obtain meaningful Brain-Score results, we would have to drastically change the evaluation pipeline before evaluating our model, and then re-run all other models for fair comparison. While we agree that the question of whether low-pass filtering at test-time will improve neural alignment is very interesting, testing this hypothesis unfortunately exceeds the scope of this work. We have updated the manuscript to discuss these considerations, though.
>
> *“[Have you attempted] testing under longer exposure times?”*
>
> Thank you for the insightful suggestion. In principle this is a good idea, but evaluating EC is problematic at longer presentation times, because humans will make fewer mistakes, so we would run into the ceiling performance issues discussed in [1]. Furthermore, human error patterns themselves become more erratic, as random mistakes caused by attentional blips or motor noise dominate the errors (we have observed both effects in yet unpublished psychophysical work). Another issue would be that at presentation times in excess of 200ms, humans will make multiple saccades, moving the experiment away from the domain of perception into the domain of visual reasoning, presumably leading to different error patterns.
>
> *“[Could you] visualize intermediate optimization steps [of the learned filter]?”*
>
> Thank you for this idea. Yes, we can visualize the intermediate optimization steps and will include a figure in the Appendix showing the progression of the filter in the final version of the manuscript.
>
> *“How sensitive is the optimal $\sigma$ to model scale and patch size?”*
>
> Thanks for this interesting question, we are currently looking into this question and should have data on the robustness of the optimal $\sigma$ to model scale and patch size soon.
>
> *“Could low-pass filtering be integrated during training to recover the lost OOD accuracy?”*
>
> The idea of integrating low-pass filtering into the training diet of models to increase their alignment has indeed been explored by various prior works, for example [2], who report modest gains in error consistency, suggesting that applying the preprocessing at train time does not lead to the same gains. Intuitively, to achieve high error consistency, models need to make mistakes on those images that humans also get wrong. Thus, training on low-pass filtered images is probably counterproductive, since such models would probably not make mistakes on critical images.
>
> *“Is the Pareto frontier analysis robust to the noisy EC estimates reported by Klein et al. (2025)?”*
>
> Great observation! We did not compute confidence intervals for the pareto frontier, because it gives the absolute maximum EC that is possible at a certain accuracy. This is a fixed value, with no uncertainty: By definition, higher values (upper bounds of confidence intervals) are impossible, because no observer is better than the perfect one.
> In principle, one could compute confidence intervals for each sequence of perfect trials with the methodology by [1] (that is, via bootstrapping). But this would account for the uncertainty that comes from the selection of stimuli, so we would obtain upper bounds that exceed the maximum for the specific stimuli that were presented to humans in MvH, because bootstrapping effectively simulates that EC was computed over a different set of stimuli. We thought this subtlety might be more confusing than helpful, so we decided to just report the frontier itself. We have, however, updated the manuscript to explicitly discuss this issue.
>
> [1] Klein, T., Meyen, S., Brendel, W., Wichmann, F. A., & Meding, K. (2025). Quantifying Uncertainty in Error Consistency: Towards Reliable Behavioral Comparison of Classifiers. In NeurIPS 39
>
> [2] Jang, H., & Tong, F. (2024). Improved modeling of human vision by incorporating robustness to blur in convolutional neural networks. Nature Communications, 15(1), 1989.

---

> > ### Comment · Reviewer_5YdF · 2025-11-26
> >
> > Thank you for the answers. Looking forward to the optimal $\sigma$ to model scale and patch size.

---

### Author Response · Authors · 2025-11-21
**General Response to Reviewers**

We would like to sincerely thank all reviewers for taking the time to evaluate our manuscript and providing such detailed and insightful feedback. We are impressed with the quality and technical depth of reviews, and delighted to read that reviewers found that our work **“addresses a key discussion in machine vision”** (tqZJ), believe that the manuscript is **“well-written”** (5YdF, V1fP) and that we provide a **“powerful reinterpretation of why generative vision models appear more human-like”** (5YdF) via a **“simple, general, physiologically grounded intervention with immediate practical impact”** (mKdF).

We incorporated the following changes into the next revision of the manuscript to address shared reviewer suggestions:

- Multiple reviewers (5YdF, tqZJ) asked about low-pass filtering at train-time. This has indeed been attempted by various prior works (Vogelsang et al. 2018, Jang et al. 2024, Lu et al. 2025), but does not seem to yield higher error consistency values than our approach. We suspect that this is because training on low-pass filtered images yields un-human-like robustness: To achieve high EC, models need to make mistakes on the appropriate images, and training models to be robust to low-pass filters might prevent them from making such mistakes. This is now discussed in the manuscript.
- There was a mistake in equation 1, which several reviewers (tqZJ, V1fP) pointed out. Thank you for paying such close attention, we have now fixed the offending letter.
- A natural follow-up question to our work (asked by reviewers 5YdF, V1fP, mKdF) concerns error consistency values at longer presentation times. Unfortunately, evaluating EC is problematic in this setting, because humans will make almost no mistakes, which is a known practical problem for the metric, which will yield unstable values. Furthermore, human error patterns become less systematic, as they are dominated by random mistakes caused by attentional blips and motor noise. At presentation times in excess of 200ms, humans also make multiple saccades, moving the experiment away from the domain of perception into the domain of visual reasoning. We therefore do not think that any reliable insights can be gained from such an experiment, but we discuss this in the manuscript now.

We expect to complete some empirical experiments within the discussion period and will update the manuscript with some requested results and figures. Again, we greatly appreciate the effort you invested in our submission and hope that our responses and the changes to the manuscript will address your concerns to satisfaction.

---

### Author Response · Authors · 2025-12-04
**Final Comment to Reviewers**

Dear Reviewers,

We have now uploaded a new version of the manuscript containing the additional figures you had requested, such as an analysis of the effect of model and patch size on the optimal $\sigma$ and a progression of the optimal filter over the course of training as well as an illustration of the effects of applying the optimal filter. We find that the optimal $\sigma$ is fairly robust to model size and patch size, but tends to increase with larger models and smaller patches. We hypothesize that the effect is stronger for smaller patch sizes because these models likely have a higher affinity and capacity for learning high frequency features prior to test-time low-pass filtering.
Additionally, we have looked into the effect of high-pass filtering on error consistency as proposed by reviewer V1fP, and find that for naive box filters, model accuracy under high-pass filtering decays quite strongly, leading to much lower error consistency values than low-pass filtering. Clearly, the low frequencies constitute important features for the model.

Thank you again for the effort you invested in our work.

---

### Author Response · Authors · 2025-12-04
**Final Comment to AC**

Dear AC,

Given the special circumstances for this year’s rebuttal period, we thought it best to briefly summarize the discussion.

We were delighted to read that reviewers judged our paper so favourably, and greatly appreciated their in-depth comments and suggestions, which were of high technical quality. Our new discussion section takes into account several reviewer suggestions, such as conducting experiments at longer presentation times and comparing to neural data. While we believe neither of these to be feasible due to technical concerns, the ideas are logical and discussing them in the paper adds value.

Following reviewer suggestions, we have more accurately approximated the human contrast sensitivity function, taking into account the boxcar presentation of stimuli, as suggested by reviewer V1fP. This better approximation indeed supports our findings slightly better than our initial version. We have also conducted an ablation on high-pass filters, finding that for the OpenCLIP ViT-H-14 model, accuracy-matched high-pass filters achieve error consistency significantly lower than low-pass filters when applied to images at test time. We have added visualizations of the intermediate stages of learning the optimal filter, as well as showing the effect of the optimal filter on a sample image. While we could not confirm with reviewers in time that our additions to the manuscript are satisfactory, we are quite hopeful that this is the case. No reviewer comments were unclear to us, and we have no reason to believe that reviewers would view the paper less favourably now than they did at the beginning of the discussion period.

---

### Meta-Review · Area_Chair_yar8 · 2026-01-05

**Summary:**

The reviewers were unanimously positive about this paper (6, 6, 8, 8), stating that the results were clearly presented, with a simple and actionable result. Reviewers raised questions and concerns about the sensitivity of the blur parameter $\sigma$ to the model scale and patch size, overfitting, and the connections to human vision.

**Reviewer Concerns:**

The authors responded to all of the reviewers' concerns during the rebuttal period. They stated that they would implement several changes; however, as of now, it does not appear that the authors have uploaded a revised version with the changes that they said they would implement. It is critical that the authors implement incorporate the reviewer feedback and their responses into the camera ready version.

**Reviewer Scores:**

Given the reviewers' initial positive assessment and the authors were responsive to the reviewers concerns, I think the reviewers would have maintained their positive scores. However, there is a possibility that because the authors did not upload a revised version with the promised changes, some authors may have lowered their scores.

---

### Decision · Program_Chairs · 2026-01-26

Accept (Poster)